# Research

materials science

BCZT ceramics, sol–gel method, $Bi_2O_3$-$B_2O_3$-$SiO_2$ doping, energy storage density

**Author for correspondence:**
Lingyu Wan
e-mail: wanlingyu75@126.com

This article has been edited by the Royal Society of Chemistry, including the commissioning, peer review process and editorial aspects up to the point of acceptance.

This article has been edited by the Royal Society of Chemistry, including the commissioning, peer review process and editorial aspects up to the point of acceptance.

# Dielectric and energy storage properties of $Bi_2O_3$-$B_2O_3$-$SiO_2$ doped $Ba_{0.85}Ca_{0.15}Zr_{0.1}Ti_{0.9}O_3$ lead-free glass-ceramics

Yaohui Chen[1,2], Daihua Chen[1], Liufang Meng[3], Lingyu Wan[1], Huilu Yao[1], Junyi Zhai[1,2], Changlai Yuan[3], Devki N. Talwar[4] and Zhe Chuan Feng[1]

[1]Center on Nanoenergy Research, Laboratory of Optoelectronic Materials and Detection Technology, Guangxi Key Laboratory for the Relativistic Astrophysics, School of Physical Science and Technology, Guangxi University, Nanning 530004, People's Republic of China
[2]Beijing Key Laboratory of Micro-Nano Energy and Sensor, Beijing Institute of Nanoenergy and Nanosystems, Chinese Academy of Sciences, Beijing 100083, People's Republic of China
[3]College of Material Science and Engineering, Guilin University of Electronic Technology, Guilin 541004, People's Republic of China
[4]Department of Physics, University of North Florida, Jacksonville, FL 32224, USA

LW, 0000-0002-9718-5595

A sol–gel method is employed for preparing high quality lead-free glass-ceramic samples $(1 - x)$BCZT-$x$BBS—incorporating $Ba_{0.85}Ca_{0.15}Zr_{0.1}Ti_{0.9}O_3$ (BCZT) powder and $Bi_2O_3$-$B_2O_3$-$SiO_2$ (BBS) glass-doped additives with different values of $x$ ($x = 0$, 0.05, 0.1, 0.15). Systematic investigations are performed to comprehend the structural, dielectric and energy storage characteristics using X-ray diffraction, field-emission scanning electron microscopy, impedance and ferroelectric analyser methods. With appropriate BBS doping ($x$), many fundamental traits including breakdown strength, dielectric loss and energy storage density have shown significant improvements. Low doping-level samples $x < 0.1$ have retained the pure perovskite phase while a second glass phase appeared in samples with $x \geq$ 0.1. As the doping level ($0.1 \geq x > 0$) is increased, the average grain size decreased to become better homogeneous materials with improved breakdown energy strengths. Excessive addition of BBS ($x = 0.15$) causes negative effects on microstructures and other traits. The glass-ceramic sample 0.95BCZT-0.05BBS exhibits excellent dielectric permittivity and temperature stability, with the highest energy storage density of 0.3907 J cm$^{-3}$ at 130 kV cm$^{-1}$. These results provide good reference to develop lead-free ceramics of high energy storage density.

# 1. Introduction

With the recent developments in device miniaturization and high-power pulse technology, many dielectric materials have attracted a great deal of interest due to their necessities in microelectronics for developing high energy storage devices such as capacitors, sensors and actuators [1–4]. Lead-free ferroelectric ceramics with environmental friendliness, such as $TiO_2$-based [5–9], $Ba_{0.94}(Bi_{0.5}K_{0.5})_{0.06}Ti_{0.85}Zr_{0.15}O_3$ (BBK) [10], $CaCu_3Ti_4O_{12}$ (CCTO) [3,11] and $BaTiO_3$-based ceramics [5–9], have become the focus of research in recent years. Among these materials, the perovskite structure $BaTiO_3$ is capable of high dielectric constant, spontaneous polarization, low dielectric loss and ferroelectricity [12,13], and offered great potentials in applications of high energy storage devices. The pure $BaTiO_3$ has, however, disadvantages of having low breakdown electric field (approximately $50\,kV\,cm^{-1}$) and poor dielectric stability of temperature. These characteristics have resulted in limited applications of $BaTiO_3$ in the field of high energy storage density of dielectric materials. The energy storage traits of $BaTiO_3$-based ceramics can be improved significantly by reducing porosity, tuning grain size, by glass additives and secondary phases, etc.

To improve the energy storage capabilities of $BaTiO_3$-based ceramics, many methods have been applied including ion doping [14], glass additives [15–17], and combining binary or multi-element systems of solid solutions [18–20], etc. In order to simultaneously attain the high dielectric breakdown strength, high energy density and high dielectric constant, a glass-ceramic concept has been devised. In this approach, a high dielectric breakdown of linear dielectric (glass) material and a substance of high dielectric constant with large polarization coefficient typical of ferroelectric ceramics are combined in a nanostructured composite type. Prepared by solid phase method, the multi-element system of $Ba_{0.85}Ca_{0.15}Zr_{0.1}Ti_{0.9}O_3$ (BCZT) exhibited larger polarization coefficient and increased breakdown electric field—it revealed, however, low energy storage density [13,21,22]. By doping BCZT with Tb, Lu *et al.* could slightly improve the energy storage density of the samples, but not the breakdown electric field [22]. The effects of NaBr, NaCl, KCl, $Na_2SO_4$ and NaCl-KCl on the structural and polarization properties of BCZT revealed the decrease of both residual polarization and breakdown electric fields [23]. By adding $CaO$-$B_2O_2$ glass powder into BCZT [21], Lai *et al.* improved the energy storage density; however, the breakdown electric field of the samples remained low. Khalf & Hall doped $0.546BaO$-$0.195B_2O_3$-$0.259SiO_2$ glass powder to pure BCZT [24] achieving a slight improvement in the *P-E* loops and high energy storage density—its breakdown electric field was still $50\,kV\,cm^{-1}$. Further improvements of $BaTiO_3$-based ceramics by defect/domain/phase boundary methods to achieve superior performances of the material have remained a challenge.

The aim of this work is to use a sol–gel method which is different from the conventional solid-state sintering method to prepare first the BCZT powders and then add barium borosilicate ($Bi_2O_3$-$B_2O_3$-$SiO_2$; BBS) glass as a sintering aid in producing a set of $(1-x)$BCZT-$x$BBS ceramic samples with different BBS components ($x$). The influence of BBS on the structural and electrical characteristics of BCZT ceramics are presented by employing X-ray diffraction (XRD), field-emission scanning electron microscopy (FE-SEM), precision impedance analyser and ferroelectric analyser. The ceramic samples with proper BBS component exhibit excellent energy storage density, high breakdown electric field and lower sintering temperature. The best sample of 0.95BCZT-0.05BBS ceramic has a slim *P-E* loop, high energy storage density ($0.3907\,J\,cm^{-3}$) and relatively low sintering temperature ($1175°C$). Our results showed that preparing high quality BCZT power by a sol–gel technique has effectively improved the performances of lead-free glass-ceramics of $(1-x)$BCZT-$x$BBS with appropriate compositions. Comparing with previous works by conventional solid-state sintering method [12,13,21–24], the samples we prepared have the advantages of relatively high breakdown electric field and energy storage density and relatively low sintering temperature. The improved densification behaviour in these material samples has potential benefits in high-voltage insulation and engineering dielectric energy storage devices.

# 2. Experimental

## 2.1. Preparation of materials

By selecting appropriate raw materials including $Ba(NO_3)_2$ (purity: 99.99%), $Ca(NO_3)_2 \cdot 4H_2O$ (purity: 99.99%), $ZrOCl_2 \cdot 8H_2O$ (purity: 99.99%), $C_{16}H_{36}TiO_4$ (purity: 99.99%) and $C_6H_2O_7 \cdot H_2O$ (purity: 99.99%), respectively we have used sol–gel method in preparing the $Ba_{0.85}Ca_{0.15}Zr_{0.1}Ti_{0.9}O_3$ nanocrystalline powders. The raw materials are first dissolved to obtain mixed BCZT gel with pH value adjusted approximately to 9. The BCZT solution is then stirred at $80°C$ and dried at $200°C$ for

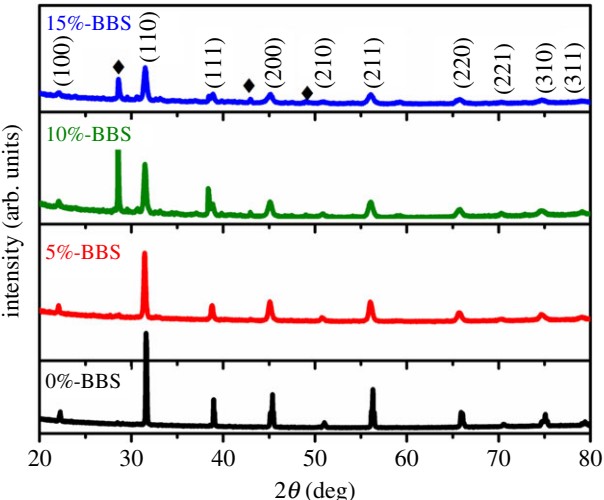

**Figure 1.** XRD patterns for the $(1-x)$BCZT-$x$BBS ceramics ($x = 0$, 0.05, 0.1, 0.15).

720 min to form desiccated gels. After BCZT powders are obtained by calcining at 1000°C for 4 h, the BBS glass as sintering additive is doped into BCZT with certain mass ratio and ground homogeneously. The resulting powders are then pressed into cylindrical pellets under a pressure of 10 MPa using 5 wt% polyvinyl alcohol (PVA) binder for 7 min. Finally, the $(1-x)$BCZT-$x$BBS samples with $x = 0$, 0.05, 0.01 and 0.15 are sintered, respectively at 1300°C, 1175°C, 1150°C and 1125°C in the atmospheric air.

## 2.2. Characterization methods

The density of $(1-x)$BCZT-$x$BBS ceramics is measured by Archimedes method. The crystal microstructures are identified by X-ray diffraction (Brüker XRD D8 discover) with Cu-Kα radiation wavelength $\lambda = 1.5418$ Å at 40 kV and 40 mA with a scanning step size of 0.02° and counting time of 0.3 s/step. Microstructures of $(1-x)$BCZT-$x$BBS ceramics are characterized using a field-emission scanning electron microscope (FE-SEM, Zeiss SUPRA 40). Ag electrodes are made on the surface of the ceramics and heated at 600°C for 20 min. By changing temperatures from 25°C to 275°C, the dielectric constant and dielectric loss of the $(1-x)$BCZT-$x$BBS ceramics are measured by a precision impedance analyser (HP, 4294 A) in a frequency range of $10^2$–$10^6$ Hz. The $P$-$E$ loops are tested using a ferroelectric analyser (TF-2000, aixACCT, Aachen, Germany) at a frequency of 10 Hz.

# 3. Result and discussion

## 3.1. XRD characterization

Figure 1 shows the crystalline structure of BCZT ceramics derived at room temperature (RT) by X-ray diffraction (Brüker XRD, D8 discover) with Cu-Kα radiation wavelength $\lambda = 1.5418$ Å. The XRD patterns of $(1-x)$BCZT-$x$BBS ceramics with different compositions ($x = 0$, 0.05, 0.1, 0.15) are displayed. It can be noted that all samples with $x < 0.1$ possess pure perovskite structure, suggesting that Ca and Zr diffuse into the BaTiO$_3$ lattice to form a solid solution. As $x \geq 0.1$, the weak peak of the second phase begins to appear, indicating that the glass starts to separate out independently. With the increase of $x$ value, the intensity of the (110) diffraction peak gradually decreases, the half-height width becomes wider, and the peak position shifts to the lower angle, which reveals that the doping of BBS glass powder changes the crystal structure of BCZT ceramics, and the lattice constant gradually increases. Doping of BBS glass into BCZT ceramics results in a lattice deformation and the changes of phase composition. As the content of BBS glass increases and the calcination temperature decreases, the electrical properties of $(1-x)$BCZT-$x$BBS ceramics may be affected due to the change of crystal structure.

## 3.2. Microstructures by SEM

Figure 2 displays SEM micrographs and grain size distribution of $(1-x)$BCZT-$x$BBS ($x = 0$, 0.05, 0.1, 0.15) ceramics were counted from SEM images by the software named 'Nano measurer'—providing characteristics of the crystallite size and agglomerate particle size of samples with different composition.

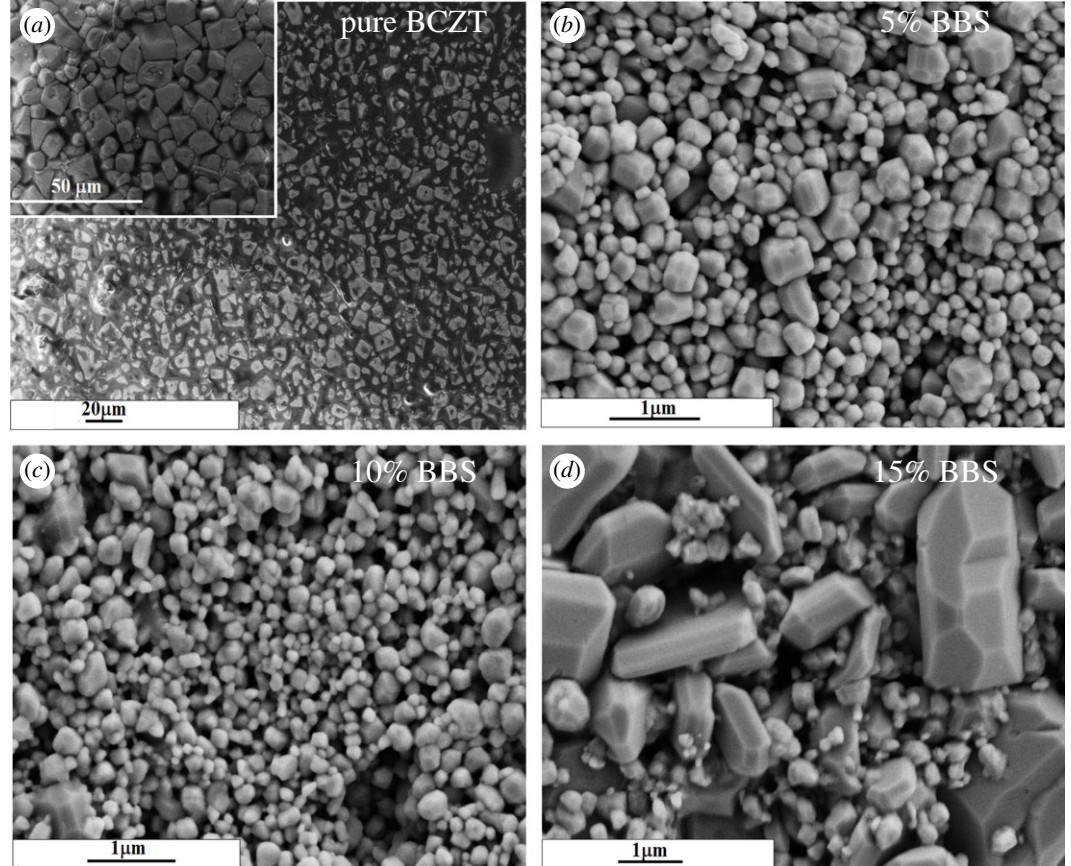

**Figure 2.** (*a*–*d*) SEM images of $(1 − x)$BCZT-$x$BBS ($x$ = 0, 0.05, 0.1, 0.15) glass ceramics.

Figure 2*a* reveals that pure BCZT ceramic with $x = 0$, exhibits clear grain boundary having mean grain size of approximately 5.76 µm. With increase of composition $x$, the average grain size of $(1 − x)$BCZT-$x$BBS ceramics decreases rapidly from 5.76 µm to approximately 207 nm for $x = 0.05$ and 196 nm for $x = 0.1$, and uniformity of the grain size increased according to figure 3. Figure 2*d* suggests that the excessive addition of BBS glass powers is detrimental to the homogenization and grain size reduction. The 0.85BCZT-0.15BBS ceramic sample shows large-size grains mixed with small-size grains, having an average size approximately 470 nm. Figures 2 and 3 have shown that the BBS glass additive has a great impact on the ceramic micromorphology.

## 3.3. Dielectric properties

Figure 4*a*,*b* illustrates the changes in the dielectric constant and dielectric loss observed at RT as a function of frequency ($10^2$–$10^6$ Hz) for the $(1 − x)$BCZT-$x$BBS ceramics. As seen in figure 4*a*, the dielectric constants of $(1 − x)$BCZT-$x$BBS ceramics decrease rapidly from 3000 to 200 as $x$ increased from 0 to 0.15. In the frequency range of $10^2$–$10^6$ Hz, the permittivity of four samples remains a good stability. In figure 4*b*, the dielectric loss of three samples with $x \leq 0.1$ has good stability of frequency. Compared with the pure BCZT, the dielectric loss of two samples with $x = 0.05$ and 0.1 is reduced by half which is kept below 0.025. For 0.85BCZT-0.15BBS sample it shows, however, unstable frequency features where the dielectric loss is high up to 0.055 at lower frequency ($10^2$–$10^4$ Hz) and decreases down to 0.025 at higher frequency ($10^5$–$10^6$ Hz). It is to be noted that the 0.95BCZT-0.05BBS sample has a low dielectric loss, a suitable dielectric constant and stable frequency dependence.

In figure 5, we have displayed the temperature dependence of dielectric constant and dielectric loss of the $(1 − x)$BCZT-$x$BBS ceramics at various frequencies. The pure BCZT ceramic exhibits a sharp phase transition peak, suggesting that it is a normal ferroelectric and has a strong relaxation behaviour. With the increase of $x$ ($x < 0.15$), the relaxation dispersion shows a gradual increase and the Curie temperature shifting to a lower temperature. At the measured frequencies, all samples exhibit good temperature stability in the range of 27°C $\leq T \leq$ 150°C. Beyond 150°C, the dielectric loss increases gradually with the heating temperature. For the same sample, the higher the frequency, the better the

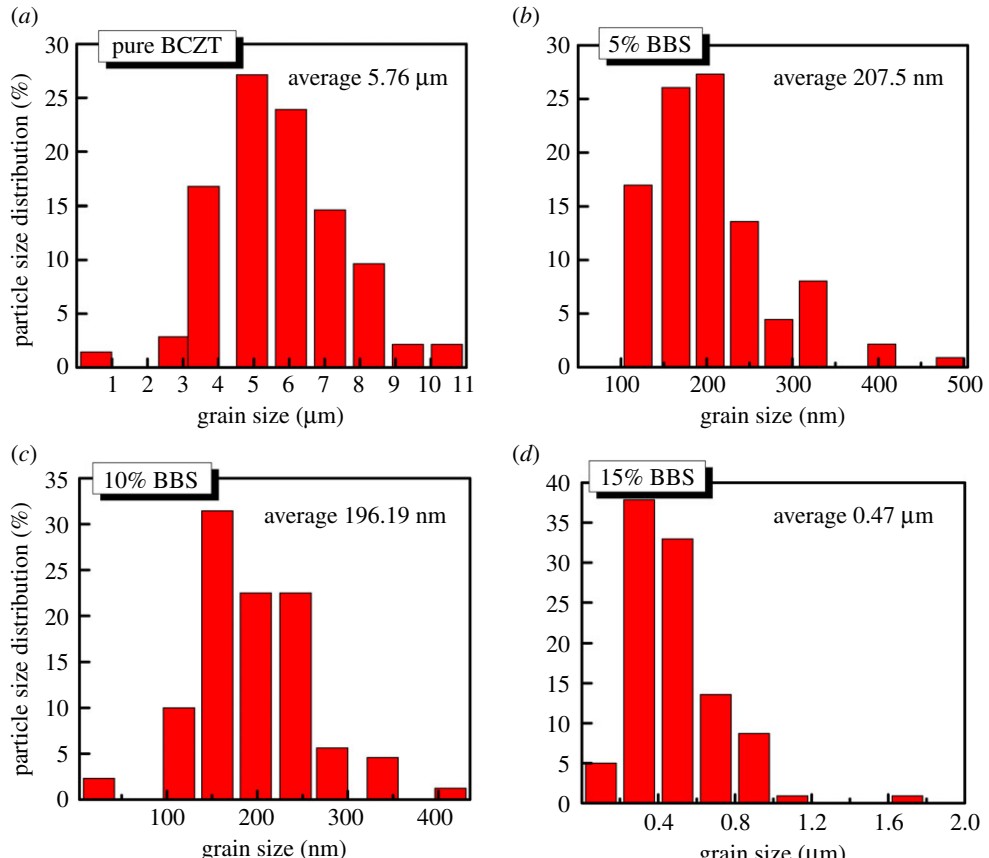

**Figure 3.** The grain size distribution of $(1 - x)$BCZT-$x$BBS glass ceramics. (a) $x = 0$, (b) $x = 0.05$, (c) $x = 0.1$, (d) $x = 0.15$.

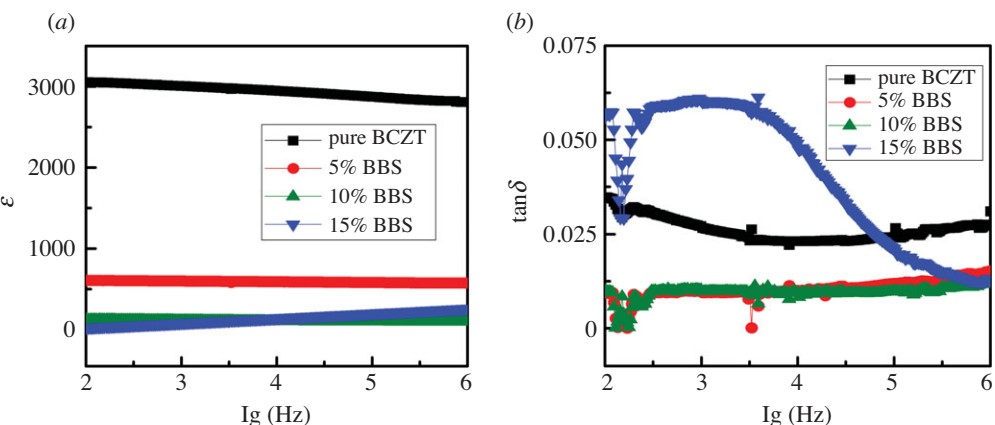

**Figure 4.** Frequency dependence of (a) dialectric constant and (b) dialectric loss for the $(1 - x)$BCZT-$x$BBS ($x = 0$, 0.05, 0.1, 0.15) glass ceramics.

temperature stability is. The permittivity and dielectric loss at high frequencies are more stable than at low frequencies in the measured temperature range. Among others, the sample with $x = 0.05$ displays a relatively good performance at temperatures below 100°C.

In typical dielectric materials, the relationships between phase transition, the dielectric permittivity and temperature obeyed the following Curie–Weiss law [25],

$$\varepsilon = \frac{C}{T - T_C}, \tag{3.1}$$

$$\Delta T = T_{CW} - T_C \tag{3.2}$$

and

$$\frac{1}{\varepsilon} - \frac{1}{\varepsilon_m} = \frac{(T - T_C)^y}{C}, \tag{3.3}$$

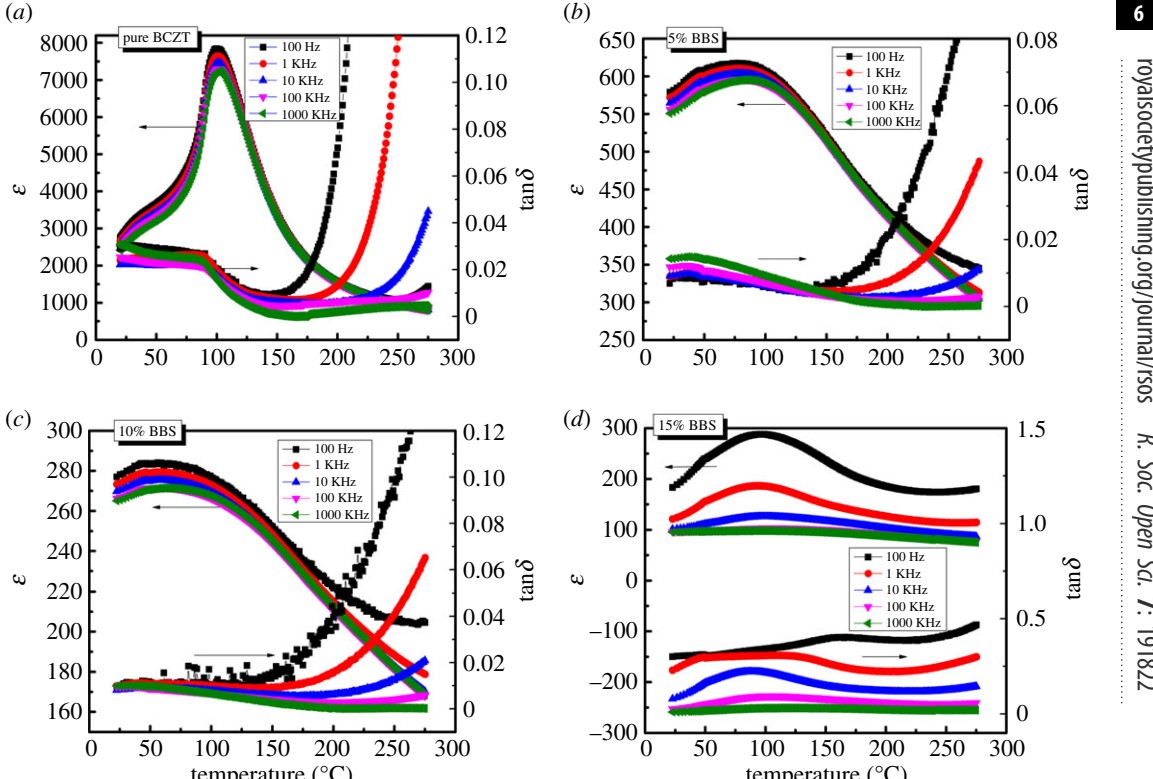

**Figure 5.** Temperature dependence of the dielectric constant and dielectric loss for the $(1-x)$BCZT-$x$BBS glass ceramics. (a) $x = 0$, (b) $x = 0.05$, (c) $x = 0.1$, (d) $x = 0.15$.

where, $T$ is the testing temperature, $T_C$ is the actual Curie temperature, $T_{CW}$ is the theoretical fit Curie temperature, $\Delta T$ is the difference between $T_{CW}$ and $T_C$, $\varepsilon$ is the permittivity and $\varepsilon_m$ is the maximum permittivity. The term $C$ is a Curie–Weiss constant, and the magnitude of $C$ is approximately $10^4$. The fitting results of the temperature dependence of dielectric content are shown in figure 6. The $(1-x)$BCZT-$x$BBS ceramics exhibit a wide phase transition peak from cubic phase to ferroelectric tetragonal phase [25,26]. The peak width is broadened, which may be due to the disorder in the microstructure [27]. The Curie temperature, $T_C$, decreases gradually and reaches the minimum value of $T_C$ of 60.1°C as $x = 0.1$, which is consistent with the trend of the grain size variation. On the contrary, the change of $\Delta T$ is exactly opposite to that of $T_C$, obtaining the maximum value at $x = 0.05$.

Figure 7 shows the comparison between experimental and simulated results of $\ln(1/\varepsilon - 1/\varepsilon_m)$ and $\ln(T - T_m)$ for $(1-x)$BCZT-$x$BBS ceramics with different $x$ values—exhibiting a linear relationship. The result demonstrates that at $x = 0.05$, the linear correlation is the best. $\gamma$ is an important parameter describing the relaxation behaviour in ferroelectric materials. When $1 \leq \gamma \leq 2$, the material behaves as a relaxed ferroelectric and has a complete diffusion phase transition. The value of $\gamma$ of samples with different BBS components are shown in table 1; all gradually decrease with increase of $x$, indicating that the increase of BBS addition has more negative influence on the relaxation behaviour of $(1-x)$BCZT-$x$BBS ceramics.

The energy storage density and efficiency of the ceramics are considered to be the most important properties in the energy storage devices. They can be calculated by using the following equations:

$$W_{\text{charge}} = \int_0^{P\text{max}} E dP, \tag{3.4}$$

$$W_{\text{discharge}} = \int_{Pr}^{P\text{max}} E dP \tag{3.5}$$

and

$$\eta = \frac{W_{\text{discharge}}}{W_{\text{charge}}} \times 100\%, \tag{3.6}$$

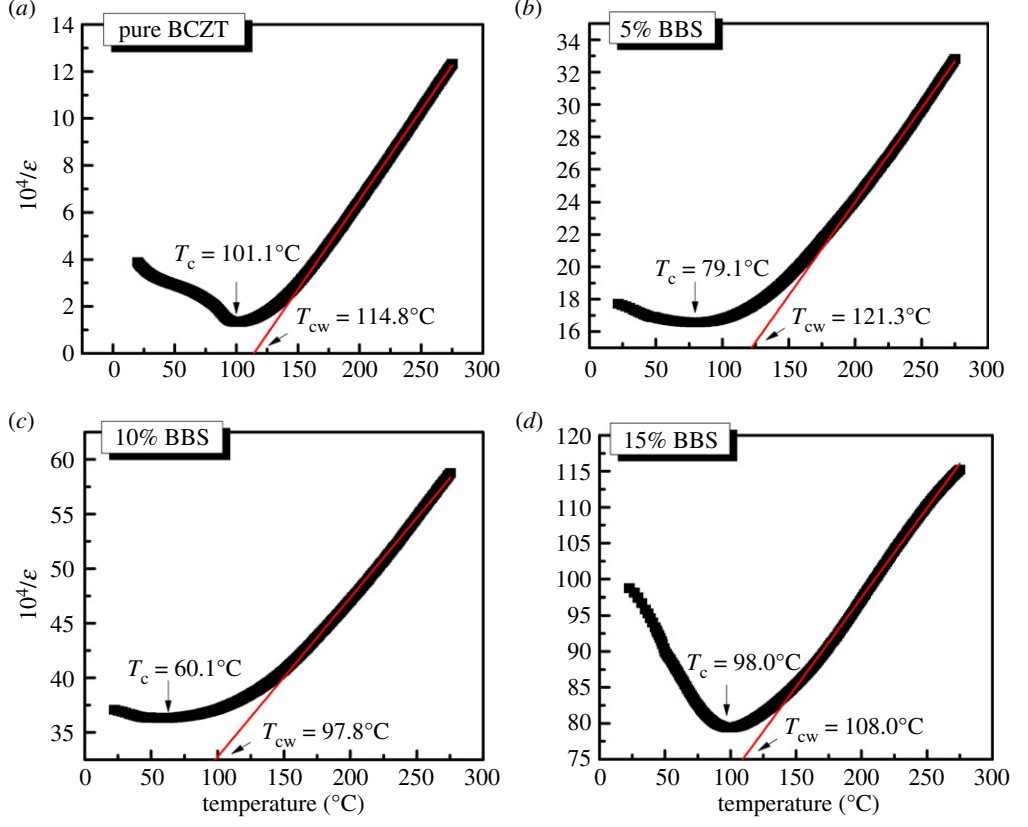

**Figure 6.** The fitted of temperature dependence of the dielectric content according to the Curie–Weiss law for the $(1-x)$BCZT-$x$BBS glass ceramics. (*a*) $x = 0$, (*b*) $x = 0.05$, (*c*) $x = 0.1$, (*d*) $x = 0.15$.

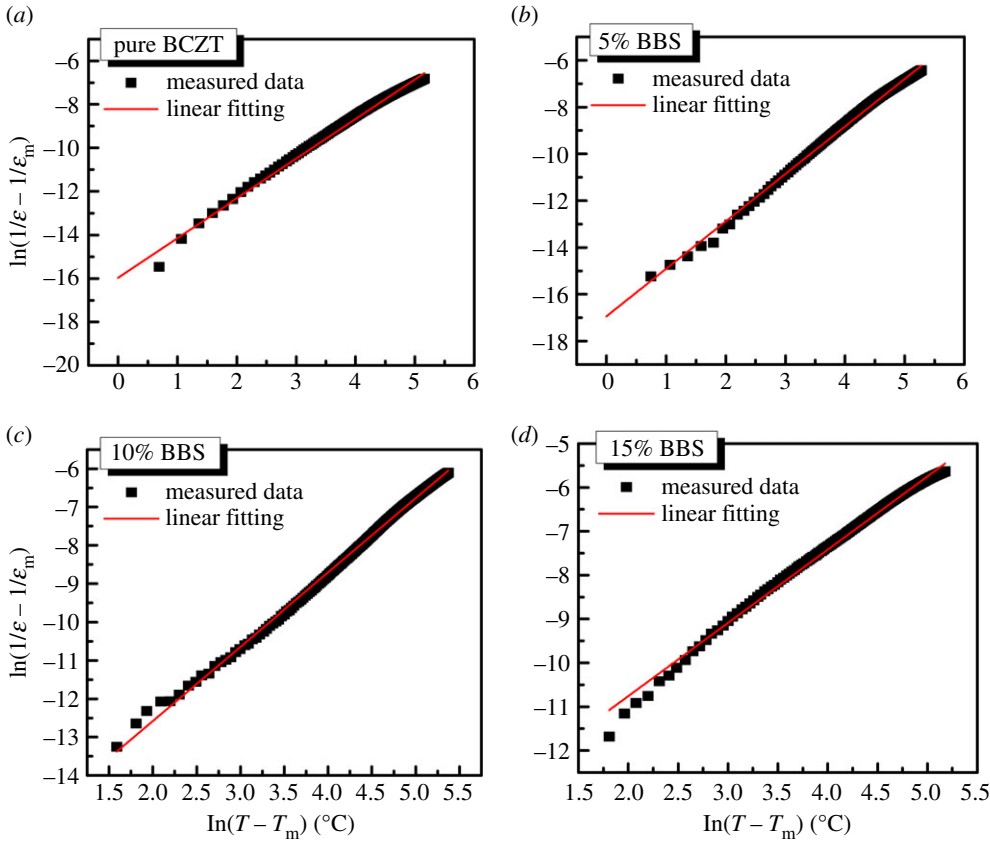

**Figure 7.** The plots of $\ln(1/\varepsilon - 1/\varepsilon_m)$ versus $\ln(T - T_m)$ at $10^6$ Hz with different values of $x$. (*a*) $x = 0$, (*b*) $x = 0.05$, (*c*) $x = 0.1$, (*d*) $x = 0.15$.

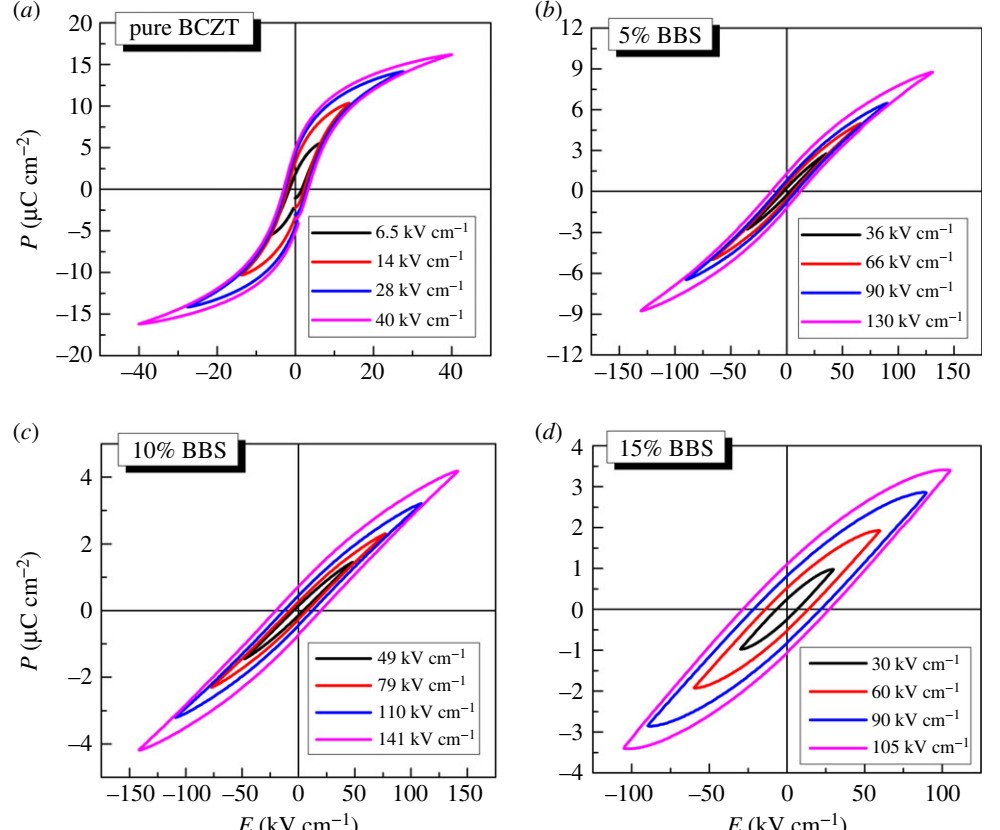

**Figure 8.** P-E loops of the $(1-x)$BCZT-$x$BBS glass ceramics with different values of $x$ at different electric field. (a) $x = 0$, (b) $x = 0.05$, (c) $x = 0.1$, (d) $x = 0.15$.

**Table 1.** The fitting parameters by the Curie–Weiss law and quadratic law for the $(1-x)$BCZT-$x$BBS ceramics with different $x$ values at $10^5$ Hz.

| sample | $T_C$ (°C) | $T_{CW}$ (°C) | $\Delta T$ (°C) | $C$ ($10^4$) | $\gamma$ |
|---|---|---|---|---|---|
| 0% BBS | 101.1 | 114.8 | 13.7 | 13.96 | 0.9771 |
| 5% BBS | 79.1 | 121.3 | 42.2 | 5.95 | 0.8644 |
| 10% BBS | 60.1 | 97.8 | 37.7 | 3.61 | 0.8197 |
| 15% BBS | 98.0 | 108.0 | 10.0 | 1.33 | 0.7192 |

where $E$ is the actual electric field intensity, $P$ is the polarization intensity, $P_r$ is the residual polarization value, $P_{max}$ is the saturated polarization value, $W_{charge}$ is the energy storage density at the time of charging, $W_{discharge}$ is the energy storage density at the time of discharge and $\eta$ is energy storage efficiency.

Figure 8 shows the P-E loops of $(1-x)$BCZT-$x$BBS ceramics at room temperature. Obviously, $(1-x)$BCZT-$x$BBS ceramics have a slim P-E loop and the better ferroelectric properties. It has a positive effect on the increase of breakdown electric field to add proper amount of BBS glass powder. The relationship between breakdown electric field and doping amount is consistent with the trend of Curie temperature variation described above. When $x = 0.1$, the maximum breakdown electric field (141 kV cm$^{-1}$) is obtained. The increase of the breakdown electric field may be attributed to the decrease of grain size. However, the addition of BBS sharply reduces the saturation polarization values.

The energy storage density is determined by P-E loops. The measured values of $P_{max}$, $W_{discharge}$, $E$ and $\eta$ for all $(1-x)$BCZT-$x$BBS ceramics are presented in table 2. Proper addition of BBS can certainly improve the energy storage density of BCZT ceramics. The $W_{discharge}$ of BCZT ceramics with BBS glass addition of $x = 0.05$ and 0.1 are obviously higher than that of pure BCZT ceramics. The maximum

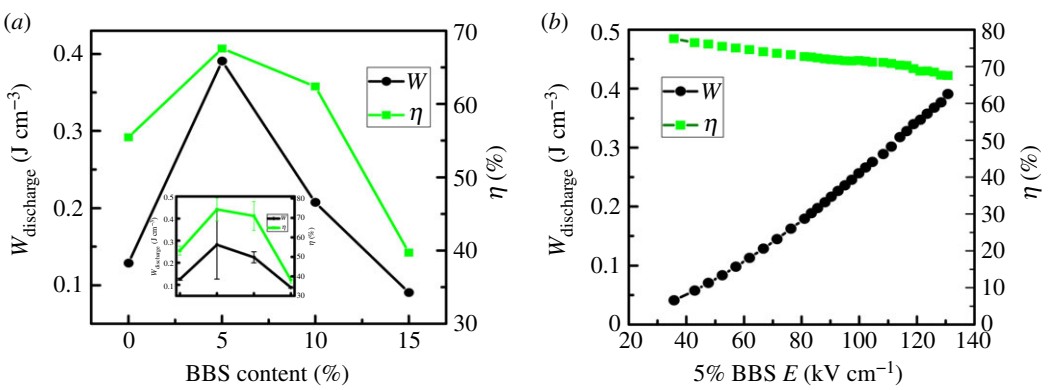

**Figure 9.** (a) Energy storage density and energy storage efficiency as a function of different values of x. (b) Energy storage density and energy storage efficiency as a function of different values of electric field when $x = 0.05$.

**Table 2.** The parameters obtained from ferroelectric properties and piezoelectric properties.

| sample | density (g cm$^{-3}$) | $W_{discharge}$ (J cm$^{-3}$) | $\eta$ (%) | $P_{max}$ (C cm$^{-1}$) | $P_r$ (C cm$^{-1}$) | $E$ (kV cm$^{-1}$) |
|---|---|---|---|---|---|---|
| 0% BBS | 5.55 | 0.1286 | 55.44 | 16.17 | 4.85 | 40 |
| 5% BBS | 4.66 | 0.3907 | 67.57 | 8.76 | 1.30 | 130 |
| 10% BBS | 4.25 | 0.2072 | 62.40 | 4.17 | 0.71 | 141 |
| 15% BBS | 4.05 | 0.0904 | 39.70 | 3.40 | 1.10 | 105 |

energy storage density reaches 0.3907 J cm$^{-3}$ at electric field of 130 kV cm$^{-1}$ as $x$ value is 0.05, which is consistent with the variation of $\Delta T$. Clearly, the energy storage efficiency and energy storage density also have the same variation tendency, reaching the maximum value at $x = 0.05$. It is found that the overall performance of 0.95BCZT-0.05BBS glass-ceramics is the best. Figure 9b gives the energy storage density and energy storage efficiency as a function of different values of electric field at $x = 0.05$. This helps us understand the dependence between $E$ and $\eta$ more clearly. It is observed that the energy storage density increases but the efficiency of energy storage deteriorates slowly with the increase of electric field. This is possibly caused by the increase of residual polarization value.

## 4. Conclusion

In this work, we have systematically investigated the effects of BBS glass additive on the dielectric, ferroelectric and energy storage properties of $(1 - x)$BCZT-$x$BBS lead-free ceramics. In samples with $x \leq 0.1$, as the BBS contents are increased, the grain sizes are obviously reduced and the microstructures of the samples become more uniform. Consequently, the relaxation diffusion degree increases gradually, and frequency and temperature stability of dielectric properties are improved. If the appropriate addition of BBS decreases, the dielectric loss and the energy storage density increases. The maximum energy storage density of 0.3907 J cm$^{-3}$ is obtained at 130 kV cm$^{-1}$ for $x = 0.05$. This lead-free 0.95BCZT-0.05BBS glass-ceramic has the advantages of high energy storage density, low energy loss and the lower sintering temperature, which has important applications in the micro high energy storage devices.

Data accessibility. All data relevant to this work are deposited at the Dryad Data Repository: https://doi.org/10.5061/dryad.jsxksn06n [28].

Authors' contributions. L.W. and Y.C. designed the study. Y.C. prepared all ceramic samples. Y.C., D.C., L.M., H.Y. and C.Y. collected and analysed the data. L.W., Y.C., J.Z., D.N.T. and Z.C.F. interpreted the results and wrote the manuscript. All authors gave final approval for publication.

Competing interests. The authors declare no competing interests.

Funding. This work was supported by the National Natural Science Foundation of China (nos. 61367004), the Guangxi Natural Science Foundation (no. 2017JJA170740y).

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
