## [Reviewer comments · Royal Society Open Science]

Review History

RSOS-191822.R0 (Original submission)

Review form: Reviewer 1

Is the manuscript scientifically sound in its present form?

No

Are the interpretations and conclusions justified by the results?

Yes

Is the language acceptable?

Yes

Do you have any ethical concerns with this paper?

No

Have you any concerns about statistical analyses in this paper?

No

Recommendation?

Accept with minor revision (please list in comments)

Comments to the Author(s)

This manuscript synthesized Ba_{0.85}Ca_{0.15}Zr_{0.1}Ti_{0.9}O₃ (BCZT) and incorporated Bi₂O₃-B₂O₃-SiO₂ (BBS) glass with various amounts. The effect of BBS on the structural and electrical properties were also investigated. I suggest accepting this manuscript after a few questions are clarified.

1. The scale bar of SEM images in Fig. 2 should be same.
2. How to get the grain size distribution? Please describe in the manuscript.

Review form: Reviewer 2

Is the manuscript scientifically sound in its present form?

No

Are the interpretations and conclusions justified by the results?

No

Is the language acceptable?

Yes

Do you have any ethical concerns with this paper?

No

Have you any concerns about statistical analyses in this paper?

No

Recommendation?

Accept with minor revision (please list in comments)

Comments to the Author(s)

The manuscript entitled "Dielectric and energy storage properties of Bi₂O₃-B₂O₃-SiO₂ doped Ba_{0.85}Ca_{0.15}Zr_{0.1}Ti_{0.9}O₃ lead-free glass ceramics" reports the study on the structural features and electrical properties of Bi₂O₃-B₂O₃-SiO₂ doped Ba_{0.85}Ca_{0.15}Zr_{0.1}Ti_{0.9}O₃ glass ceramics. The manuscript can be accepted after considering following comments.

Comment # 1: the new scientific insight is not highlighted in the introduction. The authors are suggested to include the importance of the work as compared to the activities already reported.

Comment # 2: Authors have used Ag paste based electrodes which are heated upto 600oC. At this temperature Ag may oxides and it can diffuse through the sample. Please comment.

Comment # 3: Is there any correlation between the morphology and the dielectric properties or it is solely dependent on the composition.

Comment # 4: Authors are suggested to include error bars in the Figure 9 (a).

Decision letter (RSOS-191822.R0)

30-Jan-2020

Dear Dr Wan:

Title: Dielectric and energy storage properties of Bi₂O₃-B₂O₃-SiO₂ doped Ba_{0.85}Ca_{0.15}Zr_{0.1}Ti_{0.9}O₃ lead-free glass-ceramics

Manuscript ID: RSOS-191822

Thank you for submitting the above manuscript to Royal Society Open Science. On behalf of the Editors and the Royal Society of Chemistry, I am pleased to inform you that your manuscript will be accepted for publication in Royal Society Open Science subject to minor revision in accordance with the referee suggestions. Please find the reviewers' comments at the end of this email. I apologise this has taken longer than usual.

The reviewers and handling editors have recommended publication, but also suggest some minor revisions to your manuscript. Therefore, I invite you to respond to the comments and revise your manuscript.

Because the schedule for publication is very tight, it is a condition of publication that you submit the revised version of your manuscript before 08-Feb-2020. Please note that the revision deadline will expire at 00.00am on this date. If you do not think you will be able to meet this date please let me know immediately.

Once again, thank you for submitting your manuscript to Royal Society Open Science. The chemistry content of Royal Society Open Science is published in collaboration with the Royal

Society of Chemistry. I look forward to receiving your revision. If you have any questions at all, please do not hesitate to get in touch.

Best wishes,
Dr Laura Smith
Publishing Editor, Journals

On behalf of the Subject Editor Professor Anthony Stace and the Associate Editor Professor Tobias Hertel.

RSC Associate Editor:
Comments to the Author:
(There are no comments.)

RSC Subject Editor:
Comments to the Author:
(There are no comments.)

Reviewer comments to Author:
Reviewer: 1

Comments to the Author(s)
This manuscript synthesized $\text{Ba}_{0.85}\text{Ca}_{0.15}\text{Zr}_{0.1}\text{Ti}_{0.9}\text{O}_3$ (BCZT) and incorporated Bi_2O_3 - B_2O_3 - SiO_2 (BBS) glass with various amounts. The effect of BBS on the structural and electrical properties were also investigated. I suggest accepting this manuscript after a few questions are clarified.

1. The scale bar of SEM images in Fig. 2 should be same.
2. How to get the grain size distribution? Please describe in the manuscript.

Reviewer: 2

Comments to the Author(s)
The manuscript entitled "Dielectric and energy storage properties of Bi_2O_3 - B_2O_3 - SiO_2 doped $\text{Ba}_{0.85}\text{Ca}_{0.15}\text{Zr}_{0.1}\text{Ti}_{0.9}\text{O}_3$ lead-free glass ceramics" reports the study on the structural features and electrical properties of Bi_2O_3 - B_2O_3 - SiO_2 doped $\text{Ba}_{0.85}\text{Ca}_{0.15}\text{Zr}_{0.1}\text{Ti}_{0.9}\text{O}_3$ glass ceramics. The manuscript can be accepted after considering following comments.
Comment # 1: the new scientific insight is not highlighted in the introduction. The authors are suggested to include the importance of the work as compared to the activities already reported.
Comment # 2: Authors have used Ag paste based electrodes which are heated upto 600oC. At this temperature Ag may oxides and it can diffuse through the sample. Please comment.
Comment # 3: Is there any correlation between the morphology and the dielectric properties or it is solely dependent on the composition.
Comment # 4: Authors are suggested to include error bars in the Figure 9 (a).

Author's Response to Decision Letter for (RSOS-191822.R0)

See Appendix A.

Decision letter (RSOS-191822.R1)

Dear Dr Wan:

Title: Dielectric and energy storage properties of Bi₂O₃-B₂O₃-SiO₂ doped Ba_{0.85}Ca_{0.15}Zr_{0.1}Ti_{0.9}O₃ lead-free glass-ceramics
Manuscript ID: RSOS-191822.R1

It is a pleasure to accept your manuscript in its current form for publication in Royal Society Open Science. The chemistry content of Royal Society Open Science is published in collaboration with the Royal Society of Chemistry.

On behalf of the Subject Editor Professor Anthony Stace and the Associate Editor Professor Tobias Hertel.

RSC Associate Editor
Comments to the Author:
(There are no comments.)

Reviewer(s)' Comments to Author:

Appendix A

Responses to the Referee

Reviewer #1:

1. The scale bar of SEM images in Fig. 2 should be same.

Response: We thank the referee for good suggestions. We have changed the scale bar of (b-d) SEM images to be the same. The grain size of pure BCZT ceramic is much larger than $(1-x)$ BCZT- x BBS ($x=0.05, 0.1, 0.15$) ceramics, so the scale bar of pure ceramic is different from that of $(1-x)$ BCZT- x BBS ($x=0.05, 0.1, 0.15$) ceramics.

Fig. 2. (a - d) SEM images of $(1-x)$ BCZT- x BBS ($x = 0, 0.05, 0.1, 0.15$) glass ceramics .

2. How to get the grain size distribution? Please describe in the manuscript.

Response: We thank the referee for good questions. We have been described in the manuscript. As follows:

Page 5, Microstructures by SEM section

Paragraph 2

Line 3-7: The grain size distribution was obtained according to the grain size in the SEM images by “Nano measurer” software. SEM images were imported into the “Nano measurer” software, randomly selected and measured the size of 50 grains, and the measured results were counted and analyzed to get the grain size distribution.

Reviewer #2:

Comment # 1: the new scientific insight is not highlighted in the introduction. The authors are suggested to include the importance of the work as compared to the

activities already reported.

Response: Thanks for good suggestion. The introduction has been revised in the manuscript. As follows:

Page 3, Introduction section,

Paragraph 2

Line 11-16: Compared with the work using solid-state sintering method already reported^{12,13,21-24}, our work overcomes the disadvantages of relatively small breakdown electric field, low energy storage density and relatively high sintering temperature. Our work effectively improved the breakdown electric field of the sample, significantly increased the energy storage density, and reduced the sintering temperature.

Comment # 2: Authors have used Ag paste based electrodes which are heated up to 600 °C. At this temperature Ag may oxides and it can diffuse through the sample. Please comment.

Response: We thank the referee for good questions. In this experiment , the silver slurry (containing 70% silver particles) is prepared the Ag electrodes. The silver slurry contains antioxidants, so there is no oxidation during the high-temperature heating process. In general, silver slurry contains insufficient silver, or the degree of solidification is not complete enough, and diffusion phenomenon is easy to occur. and 60%-70% of silver slurry is not prone to diffusion.

Comment # 3: Is there any correlation between the morphology and the dielectric properties or it is solely dependent on the composition.

Response: We thank the referee for pointing this. Both morphology and dielectric properties depend on the components. The morphology of ceramics has an important effect on dielectric properties. The dielectric constant decreases with the decrease of surface grain size, and the dielectric loss is the opposite.

Comment # 4: Authors are suggested to include error bars in the Figure 9 (a).

Response: We thank the referee for these valuable suggestions. We have added error bar in Fig. 9(a) in the revised manuscript.

Fig. 9. (a) Energy storage density and energy storage efficiency as a function of different values of x

All revised sentences are marked by red color and underline.

We acknowledge again for your careful reviews, kind comments and suggestions. The editor and reviewer' opinions help us to improve our work. Your efforts and time are highly appreciated.

Yours sincerely,

Lingyu WAN, Professor & Vice-Dean, Guangxi University
Email: wanlingyu75@126.com